# A Methodological Approach to Determine Sound Response Modalities to Coastal Erosion Processes in Mediterranean Andalusia (Spain)

Rosa Molina [1] , Giorgio Manno [2] , Carlo Lo Re [2] , Giorgio Anfuso [1,]* and Giuseppe Ciraolo [2]

[1] Department of Earth Sciences, Faculty of Marine and Environmental Sciences, University of Cádiz, Polígono del Río San Pedro s/n, 11510 Puerto Real, Spain; molina.gil@gmail.com

[2] Department of Engineering, University of Palermo, Viale delle Scienze, Bd. 8, 90128 Palermo, Italy; giorgio.manno@unipa.it (G.M.); carlo.lore@unipa.it (C.L.R.); giuseppe.ciraolo@unipa.it (G.C.)

* Correspondence: giorgio.anfuso@uca.es; Tel.: +34-956-016167

**Abstract:** Human occupation along coastal areas has been greatly increasing in recent decades and, in many places, human activities and infrastructures are threatened by erosion processes that can produce relevant economic and human losses. In order to reduce such impacts and design sound management strategies, which can range from the "no action" to the "protection" option, coastal managers need to know the intrinsic coastal sensitivity and the potential vulnerability and value of land uses. In this paper, in a first step, coastal sensitivity was determined by calculating the following: (i) the spatial distribution at the coast of the wave forcing obtained by using the ERA5 wave dataset and defined as the energy associated with the 50-year return period storm. Two storm conditions were considered, that is, one for the eastern and one for the western parts of the Andalusia Mediterranean coast, respectively, characterized by a height of 8.64–7.86 m and 4.85–4.68 m and (ii) the existence of a buffer zone, namely the dry beach width expressed as a multiple of the 20-year predicted shoreline position that was calculated using a dataset of aerial photographs covering a time span from 1956 to 2016. Coastal sensitivity values were divided into five classes with class 1 indicating the lowest sensitivity (i.e., the presence of a wide buffer zone associated with low wave energy flux values) and class 5 the highest sensitivity (i.e., a narrow buffer zone associated with very high wave energy flux values). In a second step, land uses were obtained from the official Land Use Map of the Andalusia Region, based on the results of the "Coordination of Information on the Environment" (CORINE) European Project. Such uses were divided into five classes from class 1 including natural areas (typologies "A" and "B" of the CORINE Project) to class 5 including very capital land uses (typologies "E1" and "E2"). In a third step, information concerning coastal sensitivity and land uses was crossed to determine the best mitigation strategies to cope with erosion processes. The "no action" option was observed at the westernmost area of Cádiz Province and at some areas from the west coast of Almería Province, where both coastal sensitivity and land use classes show low values; the "adaptation" option was recorded along more than one half of the coast studied, essentially at natural areas with high sensitivity and at urbanized areas with low sensitivity; and the "protection" option was observed especially at some areas from the center and eastern part of Málaga Province and at the easternmost areas of Almería Province, where both coastal sensitivity and land use classes presented high values.

**Keywords:** coastal trend; wave energy; beach width; land use; mitigation

## 1. Introduction

Ocean coastlines are highly dynamic and changing environments since they show great temporal and spatial variability in response to the action of different and complex coastal processes essentially linked to waves and currents [1]. Often, beach erosion/accretion cycles are recorded at an inter-annual time scale and are related to seasonal wave climate variations due to temporal and spatial distributions of high latitude storms and hurricanes/typhoons [2–6]. Erosion is observed after storm events, at high latitudes recorded during winter months, but beach recovery takes place during fair weather conditions, which is known as "seasonal" beach behavior [7,8]. In this case, erosion processes represent a threat since they can locally menace human structures/activities, but the following natural beach recovery guaranties the reformation of a wide beach and its associated protection function and tourist use.

Different is the case in which coastal erosion is the result of a long, decadal trend [9–11] due to the impact of large storms and tsunamis [12–14], sea level rise, and variations in sediment supply, which is linked to river contributions and longshore and cross-shore current supplies. River contributions are linked to variations in rainfalls, changes in land use (e.g., soil erosion increases when forests and grasslands are converted into farm fields and pastures) and the construction of dams and the channelization of river banks [15,16]. Longshore and cross-shore supplies record variations because of changes in wave climate and current pathways [17,18] or the accumulation updrift of human structures (e.g., ports, groins, etc. [16]). In this case, erosion processes produce important retreat (with no or partial associated recovery) at natural places reflected by overwash and/or beach and dune erosion [11], and damages at the location of human activities or infrastructures in urbanized coastal sectors [9], that is, storms become natural hazards [19].

In order to reduce such impacts and associated economic and human losses, coastal managers need to know the following: (i) the sensitivity of natural coastal sectors, which is related to wave energy, beach characteristics/evolution, and sea level trend [20,21] as well as (ii) the potential vulnerability and economic value of the urbanized sectors [9,22].

Wave energy characteristics, especially the spatial and temporal distributions and the periodicity of most energetic events, are the forcing agents that drive morphological beach changes and determine erosive/accumulation processes [23,24] that also depend on beach dynamics and characteristics (i.e., beach morphodynamic state, width, presence of dune ridges, etc. [9]).

Human activity and infrastructure vulnerability depend on the distance from the shoreline and their typology [9,25]. Following different criteria, vulnerability maps have been obtained for numerous coastal areas around the world through the use of computer-assisted multivariate analysis, numerical models, and Geographical Information Systems (GIS), with pioneer investigations carried out by Gornitz [26] and Gornitz et al. [27] in the USA, by Cooper and McLaughlin [28], McLaughlin et al. [22], and the Committee on Climate Change [29] in UK, and by Dodds et al [30] in Malta and Mallorca.

Once coastal sensitivity/vulnerability is determined, the next step for coastal managers is to decide which are the most appropriate mitigation strategies that include "do nothing", "adaptation", and "protection" [31,32]. Do nothing is the "no action" option (i.e., the decision to not defend properties at risk and/or the abandonment of the current defense line). Adaptation includes accommodation (i.e., the modification of human infrastructures and land uses; e.g., an agricultural area may be replaced by a salt marsh) and relocation (i.e., the landward displacement of human activities/uses). Protection is the "hold the line" option (i.e., the defense and maintenance of the current shoreline position using hard armoring structures or beach nourishment [29]). The selection of the sound management strategy is based on the knowledge of the erosion processes (magnitudes and causes) and funding/legislation (The Shoreline Management Guide, http://www.eurosion.org/project/eurosion_en.pdf, accessed on 3 November 2019). Economic considerations are based on a cost–benefit analysis approach [23] or on action/reaction criteria [9].

The present paper presents a methodological approximation to determine coastal sensitivity and sound response actions to long-term erosion processes that affect sandy sectors along the Mediterranean

coast of Andalusia (Spain), an area that records an important flow of tourists associated with the "Sun, Sea, and Sand (3S) market" [30], which represents a relevant economic resource for the region.

In a first step, coastal sensitivity was determined by calculating the spatial distribution of wave forcing and the existence of a buffer zone, namely the dry beach width expressed as a multiple of the 20-year predicted shoreline position. In a second step, land uses were obtained from the official web page of the Regional Administration of Andalusia. In a third step, coastal sensitivity was related to land uses in order to determine the best mitigation strategies to cope with erosion processes. The methodology proposed in this paper represents a valuable tool for coastal managers at a regional scale and can be easily applied in different coastal areas around the world where basic information on the delineated parameters is available. Once information is obtained at the regional scale, further studies and investigations can be devoted to determine best adaptation options and specific defense methods at a small spatial scale. In this paper, sea level change has not been taken into account because this investigation is devoted to determine human infrastructure vulnerability and sound response options in the next few decades, that is, at a time span for which sea level variations are usually not relevant according to Komar [1].

## 2. Study Area

The Mediterranean coast of Andalusia, located in SW Spain, extends from the Gibraltar Strait to the Murcia Region, with a length of ca. 546 km (ca. 196 km composed of cliffed and ca. 350 km of sand sectors) administratively belonging to the provinces of Cádiz, Málaga, Granada, and Almería. It has an E-W prevailing rectilinear outline with two NE-SW oriented and easterly facing sectors, that is, at the Almeria and Gibraltar areas (Figure 1).

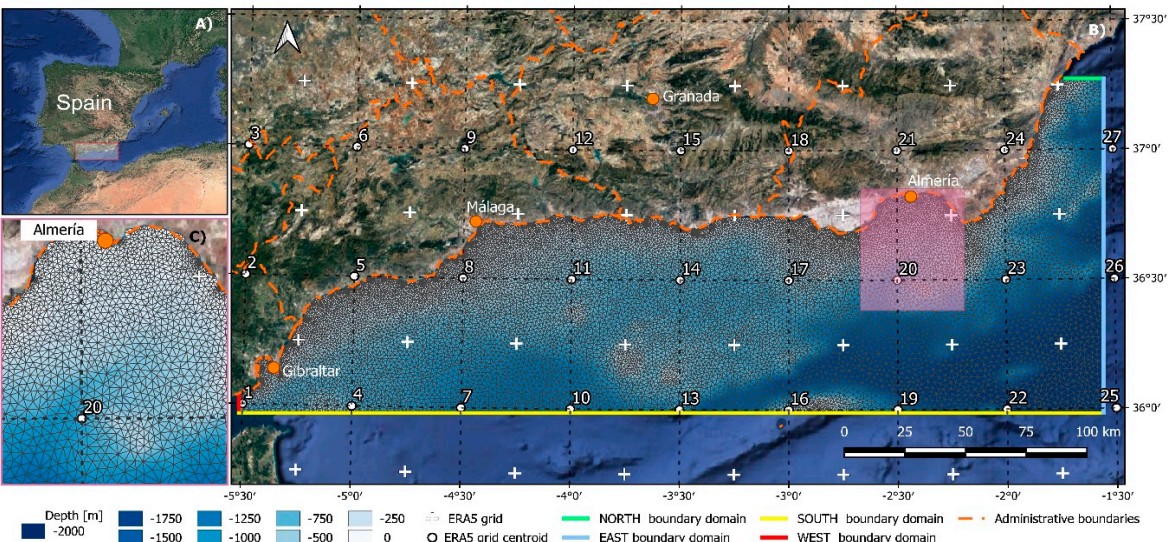

**Figure 1.** (**A**) General view of the numerical domain area used in this paper. (**B**) Numerical domain and triangular mesh sides; white dots represent the ERA5 data grid points and white crosses represent the corner of each grid. (**C**) Detail of the Delaunay triangulation used for the Gulf of Almeria.

The Betic Range, a tectonically active mountain chain that reaches high elevations close to the coast, strongly controls coastal orography. The Guadiaro, Guadalhorce, Guadalfeo, Adra, and Andarax are the most important rivers and "ramblas" (a seasonal stream), draining the chain and favoring the formation of small coastal plains developed at the foot of the chain. Especially under episodic heavy rainfalls, linked to the semiarid climate, fluvial sands and gravels are transported by rivers and ramblas to the coast, constituting important sediment supplies to the beach system. River basin regulation plans in the last decades have brought the construction of dams and reservoirs that have

significantly limited sediment supplies to the coast, thereby exacerbating coastal retreat, especially at many deltas [33–36].

Beaches, which usually show medium to coarse dark sands and/or pebbles and intermediate to reflective morphodynamic states, are found along the provinces of Cádiz, Málaga, and Granada [33,37]. Meanwhile, more dissipative beaches, composed of fine/medium quartz rich sand, are observed in Almeria Province [38,39]. Beaches are often interrupted by rocky sectors and headlands that give rise to pocket beaches ("calas") of different sizes and relevant scenic value [38].

The coastal area is a micro-tidal semidiurnal environment (tidal range < 20 cm) [33] exposed to winds blowing from SE to SW with minimum and maximum velocities ranging from 0.4 to 9.0 m/s. The Gibraltar Strait area is affected by eastern storms, whereas the Málaga and Almeria areas are exposed both to western and eastern storms [40]. The predominance of wave climate shows a seasonal trend with storm events essentially recorded during November–March, that is, the winter season [40,41]. The prevailing littoral drift, linked to both coastal orientation and predominantly easterly winds and associated storm waves, is westward directed [40,41]. An opposite littoral drift, observed at specific places, is associated with sea and swell waves entering from the Gibraltar Strait [33,40].

From an administrative point of view, the coast belongs to the provinces of Cadiz, Malaga, Granada, and Almeria, with the large coastal towns of Malaga (with >500,000 inhabitants), Almeria (with ca. 200,000inhabitants), and the tourist towns along the western part of Costa del Sol area, namely Marbella (with 150,000inhabitants), Fuengirola (80,000), and Torremolinos (70,000). According to DGPC [42], coastal uses along Andalusia, expressed in percentage, include tourist/recreational uses (41.1%, which are especially observed in Malaga Province), natural (4.7%, essentially at Almeria and Cadiz provinces), port/commercial (2.5%, with main commercial ports located at Almeria, Algeciras, Cadiz, and Malaga and several marinas at Costa del Sol), industrial (2.3%), and fishing (0.9); the remaining coast has no defined use.

With respect to sea level trend, Criado-Aldeanueva et al. [43] analyzed a 16-year time set of sea level data for the Mediterranean Sea obtained by means of satellite radar images and observed that the most relevant sea level rise took place in the Levantine basin south of Crete with values up to 10 ± 1 mm/year. Some other rising spots were recorded in the Adriatic and Alboran Seas with more moderate positive trends. Tsimplis et al. [44], based on tidal gauge datasets of 10 to 58 years, gave a negative trend at the Gibraltar Strait area (−1.21 m/yr) and a positive trend at Málaga and Almería (4.04 and 0.28 mm/yr, respectively). Lastly, Puertos del Estado [45] presented data on sea level trends based on tidal gauge records including dataset periods ranging from 8 to 25 years. Close to the Gibraltar Strait and at Almeria, sea level trends recorded a negative trend (−0.345 and −0.046 cm/yr, respectively); meanwhile, at Motril and Málaga, they presented a positive one (0.181 and 0.231 cm/yr, respectively).

## 3. Methodology

For this study, sound mitigation actions to alleviate coastal erosion were proposed by combining coastal sensitivity with land use information (Figure 2). Coastal sensitivity was obtained by combining the main forcing agent, expressed using the longshore distribution of the wave energy flux, with the extent of the buffer zone (i.e., the presence of a beach expressed considering dry beach width as a function of the 20-year predicted shoreline position), and land use was obtained by consulting existing official information, namely the SIOSE Project Land Use Map of the Regional Administration of Andalusia (Spain) (http://www.juntadeandalucia.es, accessed on 4 November 2019).

### 3.1. Coastal Forcing

With respect to wave forcing, wave climate analysis was carried out using the ERA5 wave dataset modeled by the European Centre for Medium-range Weather Forecasts ECMWF within the framework of the homonym project, using the WAve Model (WAM) that is a numerical model based on the energy balance equation able to reconstruct wave climate [46] (https://www.ecmwf.int, accessed on 4 November 2019). The data cover the whole Earth on a 30 km grid and has a time interval of 40 years,

from 1979 to 2019, which is representative of potential trends of increasing wave height and/or the presence of climate-controlled cycles [6].

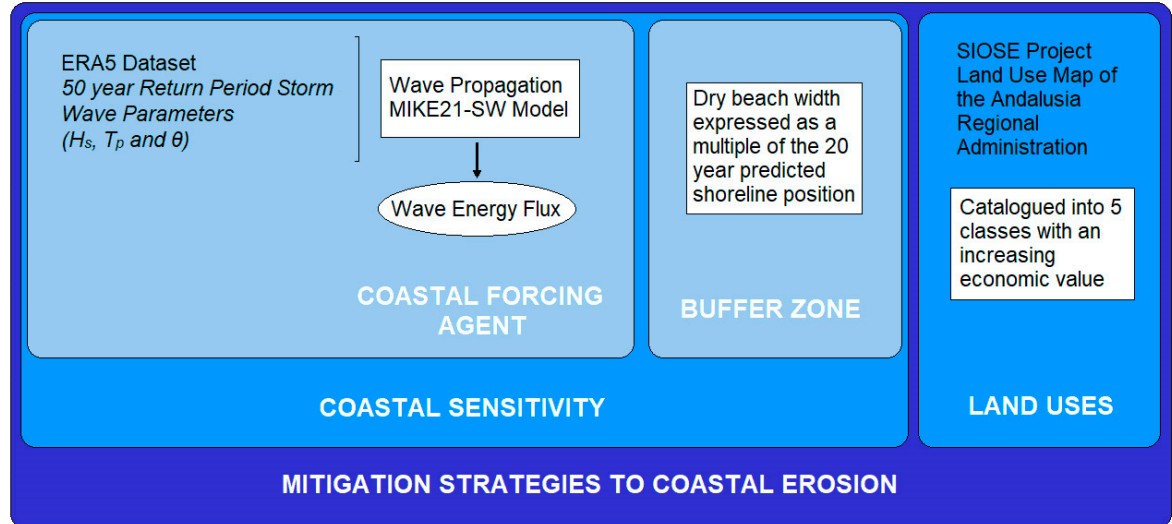

**Figure 2.** Summary of the methodology used in this work.

This investigation used 27 points among the available grid points from the ERA5 dataset along the Mediterranean coast of Andalusia (Spain, Figure 1). The dataset, analyzed using MATLAB scripts, included the significant wave height ($H_s$), the peak period ($T_p$), and the mean wave direction ($\theta$) at each point of the grid. Once the offshore wave parameters were known, a statistical analysis of sea storm (directional and omnidirectional) was carried out. In this paper, each single storm was defined as "a sequence of sea states in which $H_s$ exceeds a fixed threshold $h_{crit}$, and does not fall below this threshold for a continuous time interval greater than 12 h" [47]. The $h_{crit}$ value was assumed equal to 1.5 m and the linear deep-water wave theory allowed to calculate the storm-associated energy flux [40,47].

The data were divided into 16 main directions; for each direction it was possible to fit the extreme value distribution related to each point of the ERA5 grid. For each direction and for the whole dataset, the best fitting extreme distribution function was chosen among the generalized extreme value (GEV), Gumbel, and Weibull distributions. The goodness of fit of these distributions was assessed using the well-known Kolmogorov–Smirnov test. The result of this analysis was that the GEV distribution had the best fitting performance for all the datasets.

The GEV distribution is a family of continuous probability distributions developed within extreme value theory to combine the Gumbel, Fréchet, and Weibull families, also known as type I, II, and III extreme value distributions. The return period was calculated by the following expression:

$$T_r = \frac{1}{\lambda \cdot P(H_s, \theta)} \tag{1}$$

where $\lambda$ is the mean number of storms in a year and $p$ is the exceedance probability given by the GEV equation in which the parameters were calculated using the maximum likelihood estimation (MLE) method.

In order to assess storm effects along the service life of coastal structures, a return period of 50 years was used because the aim of the research was focused on mid-term period damages. Only the more representative directions were used to force the model at the boundaries.

The wave propagation from offshore to nearshore was performed using the well-known MIKE21-SW model [48,49]. This model was used with the directionally decoupled parametric formulation of the wave action balance equation. Moreover, the runs were performed in the non-stationary mode reproducing a storm of 48 h.

The GEBCO 30 arc second bathymetric grid (General Bathymetric Chart of the Ocean) was used to calculate the depth of mesh points [50]. The mesh was created by means of a Delaunay triangulation and the triangle size was determined using a density function taking into account both the local depth and the bottom slope (Figure 1), obtaining a great detail near the coastline. The mesh used in the wave model was made of 30,906 triangular elements and 16,151 nodes. The smallest length of the elements was 14.93 m (near the coastline), and the maximum size was 1988 km in deep water. The smallest and the largest angles of the triangles were ca. 26° and 60°, respectively. The outputs of the numerical model are wave parameters ($H_s$, $T_p$, and $\theta$) and the energy flux at each numerical node of the domain. In order to extract the energy flux, all the mesh elements that belonged to the 10 m depth isobath were selected. In order to assess wave effects in the nearshore, the energy flux parameter was used because it takes into account both wave height and length during a storm [51]. The energy flux represents the power of the storm per unit of wavelength.

Following McLaughlin et al. [22] and Rangel-Buitrago and Anfuso [9], the forcing agent, which in this paper is the wave energy flux distribution along the coastline, was divided into five classes by "the natural breaks" function [52] according to its impact on the coast, from "Very Low" (Class 1) to "Very High" (Class 5, Table 1).

**Table 1.** Classes of wave energy flux (kw/m), the colors are related to the intensity of possible erosion effects.

| 1 | 2 | 3 | 4 | 5 |
|---|---|---|---|---|
| **Very Low** | **Low** | **Medium** | **High** | **Very High** |
| 0.0–7.65 | 7.65–15.45 | 15.45–40.09 | 40.09–79.75 | 79.75–146.12 |

### 3.2. Buffer Zone Assessment

The dry beach width represents a buffer zone to storm impact and hence it must be taken into consideration in coastal sensitivity determination [9]. Beach slope and elevation are also important factors in beach behavior and coastal protection function, but they presented a great temporal and spatial variability as observed on the Atlantic side of Andalusia [5]. Furthermore, no detailed data are available for the whole length of the coast investigated, hence beach width was used as an indicator of beach protection function. Beach width was considered as the distance between the most recent available shoreline and the landward limit of the beach, coinciding with the dune foot or human structures (promenades, seawalls, etc.). Both lines (i.e., shoreline and landward beach limit) were mapped on the most recent available aerial photos, namely the 2016 orthophotographs of the Regional Administration of Andalusia (http://www.juntadeandalucia.es, accessed on 25 October 2019). Since dry beach width can have a considerable range from coast to coast, it was expressed as a multiple of the 20-year predicted shoreline position [53], using the methodology proposed by Rangel-Buitrago and Anfuso [9]. This parameter reflects coastal evolution and hence coastal response to erosion processes that is linked to beach characteristics such as beach width, elevation, and slope/morphodynamic state [5]. The 20-year predicted shoreline position represents the area subject to erosion in the next 20 years and is extended landward from the shoreline for a distance equal to 20 times the average, annual erosion rate for the site calculated for medium- (10–60 years) or long-term time spans (>60 years), [54]. In this paper, the 20-year predicted shoreline position was selected as a reliable indicator of future shoreline trends because, according to Leatherman et al. [55], in order to be reliable, the predicted time span has to be less than 1/2 of the total period considered (i.e., 60 years).

Evolution rates are usually available from previous studies and are obtained by means of aerial photographs [56]. Molina et al. [36] used aerial orthophotographs from 1956, 1977, 2001, 2010, and 2016 to reconstruct and quantify shoreline evolution, and these data were used in this paper. The orthophotos were obtained by the Regional Government (Junta de Andalucía, (http://www.juntadeandalucia.es, accessed on 25 October 2019), and all information was presented in the metric Projected Coordinate System WGS84, UTM zones 29 N and 30 N [36]. The five shorelines used were mapped using the

ArcMap application from ArcGIS Desktop, Release 10. Redlands, CA: Environmental Systems Research Institute, and the shoreline position was defined as "the water line at the time of the photo" [57,58] because of the micro-tidal nature of the studied coast. Corrections concerning shoreline position were carried out taking into account the photos' own characteristics and the digitalizing process [59], that is, digitalizing error, accuracy linked to pixel size, ortho-rectification error, image co-registration error as well as shoreline definition and position determination, namely wave run-up and tidal conditions [10,36].

Because of the accuracy of the method used, the evolution of cliffed sectors, which have an overall length of ca. 196 out of 546 km, was not quantified and hence such sectors were considered stable at the scale of this investigation, that is, changes were too small to be detected by means of aerial orthophotographs. Hence, the evolution of a total amount of 284.95 km of beach sectors was obtained by drawing 10,073 shore-normal transects, with a spacing fixed at 25 m [36]. Rates of change between shorelines were computed using the DSAS extension of ArcGIS [60,61] by calculating the weighted linear regression (WLR), which considers all used shorelines.

Hence, according to Rangel-Buitrago and Anfuso [9], the buffer zone (i.e., the dry beach width expressed as a multiple of the 20-year predicted shoreline position) was assessed in correspondence with each one of the 10,073 shore-normal transects and expressed into five classes according to its relative width (Table 2).

**Table 2.** Buffer zone classes expressed according to the dry beach width and the 20-year predicted shoreline position the color scale indicates the intensity of exposure i.e. a large buffer zone shows a low exposure.

| 1 | 2 | 3 | 4 | 5 |
|---|---|---|---|---|
| **Very Wide** | **Wide** | **Medium** | **Narrow** | **Very Narrow** |
| Dry beach width ≥5 times the 20-year shoreline position | 4 times the 20-year shoreline position | 3 times the 20-year shoreline position | 2 times the 20-year shoreline position | ≤1 time the 20-year shoreline position |

### 3.3. Coastal Sensitivity

Coastal sensitivity was obtained by calculating the arithmetic average value of forcing and buffer zone classes and was expressed into five classes. As this study was carried out at a large, regional scale, coastal sensitivity was obtained averaging buffer zone values along 200 m long sectors within each uniform land use unit.

### 3.4. Land Use

Land use categories, presented in the SIOSE Project Land Use Map of the Andalusia Regional Administration (http://www.juntadeandalucia.es, accessed on 4 November 2019), were used in this paper. These categories were based on the results of the European project "Coordination of Information on the Environment" (CORINE, http://www.eea.europa.eu, accessed on 12 October 2019) and, in this investigation, were geo-referenced and catalogued into five classes with an increasing economic value from class A to E (Table 3).

**Table 3.** Land use economic value according to the Land Use Map of the Andalusia Regional Administration. Codes of land uses (i.e., "A", "B", "C1", "C2", "D2") are those used by the "Coordination of Information on the Environment" (CORINE) Project.

| A | B | C | D | E |
|---|---|---|---|---|
| **Very Low** | **Low** | **Medium** | **High** | **Very High** |
| Open spaces with little or no vegetation **(A)** Scrub and/or herbaceous vegetation associations **(B)** | Forests **(C1)** Wetlands and water bodies **(C2)** | Agricultural areas **(D2)** Artificial, non-agricultural vegetated areas **(D3)** | Discontinuous urban fabric **(D1)** | Continuous urban fabric **(E1)** Industrial, commercial, and transport units and mine, dump, and construction sites **(E2)** |

*3.5. Mitigation Strategies to Coastal Erosion*

According to Pranzini et al. [31] and Williams et al. [32], there are several mitigation or response strategies to counteract coastal erosion processes that range from "do nothing" and "adaptation" to "protection". In this paper, three main options were considered: (1), "no action" (i.e., no action is required because coastal area has land uses of low economic value that are at null or low risk); (2) "adaptation", which includes "accommodation" (i.e., the modification of existing infrastructures and/or the change in land use) and "relocation" (i.e., the landward movement of infrastructures); and (3) "protection" (i.e., the establishment of hard structures as groins, breakwaters, etc., as well as the execution of nourishment works, when land use at risk is of great economic value).

## 4. Results

*4.1. Wave Forcing*

The open boundary of the domain, divided into four segments, namely North, East, South, and West, was used to force the model (Figure 1). For each of these boundary segments, a point from the ERA5 re-analysis grid, namely No. 1, 13, 26, and 27, was chosen. For each ERA5 prediction point, directional roses of significant wave height were extracted (Figure 2). Only two main directions (i.e., ENE and WSW) recorded a significantly large number of storms (Figure 3, Table 4). Hence, the $H_s$ wave height, considering a storm with a 50-year return period, was calculated at each point and for each direction (Table 4).

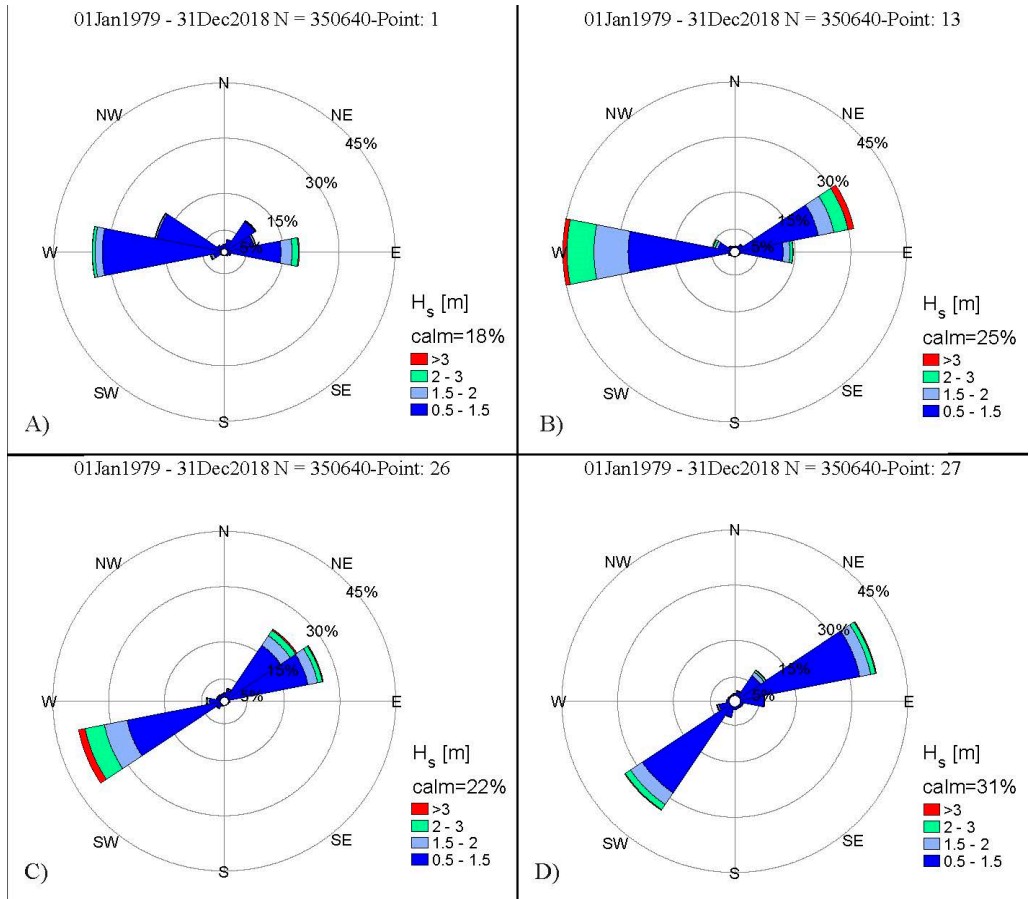

**Figure 3.** Significant wave height roses obtained considering the time span from 1 January 1979 to 31 December 2018 for ERA5 point nos. 1 (**A**), 13 (**B**), 26 (**C**), and 27 (**D**).

**Table 4.** $H_s$ and $T_p$ wave parameters in the lateral boundary of the numerical domain for $T_r$ = 50 years.

| Points | Coordinates | θ (°) | Cardinal Points | $H_s$ (m) | $T_p$ (s) |
|--------|-------------|-------|-----------------|-----------|-----------|
| 1 | 36° N 5°30′ E | 247.5 | WSW | 4.68 | 9.14 |
| 13 | 36° N 3°30′ E | 247.5 | WSW | 4.85 | 6.44 |
| 26 | 36°30′ N 1°30′ E | 67.5 | ENE | 8.64 | 9.36 |
| 27 | 37° N 1°30′ E | 67.5 | ENE | 7.86 | 10.25 |

The significant wave height value ($H_s$) with a 50-year return period was calculated, for each point and for each direction of approximation (Table 4).

Using the wave parameters presented in Table 4, two storm conditions were propagated. Significant wave height distribution at the end of the simulation processes was shown for two storms approaching from ENE and WSW (Figure 4A,B). Combining significant wave height and the associated peak period, the wave power distribution per unit of wavelength (i.e., the energy flux) was obtained for the whole domain and presented for the Andalusia coastline (Figure 5).

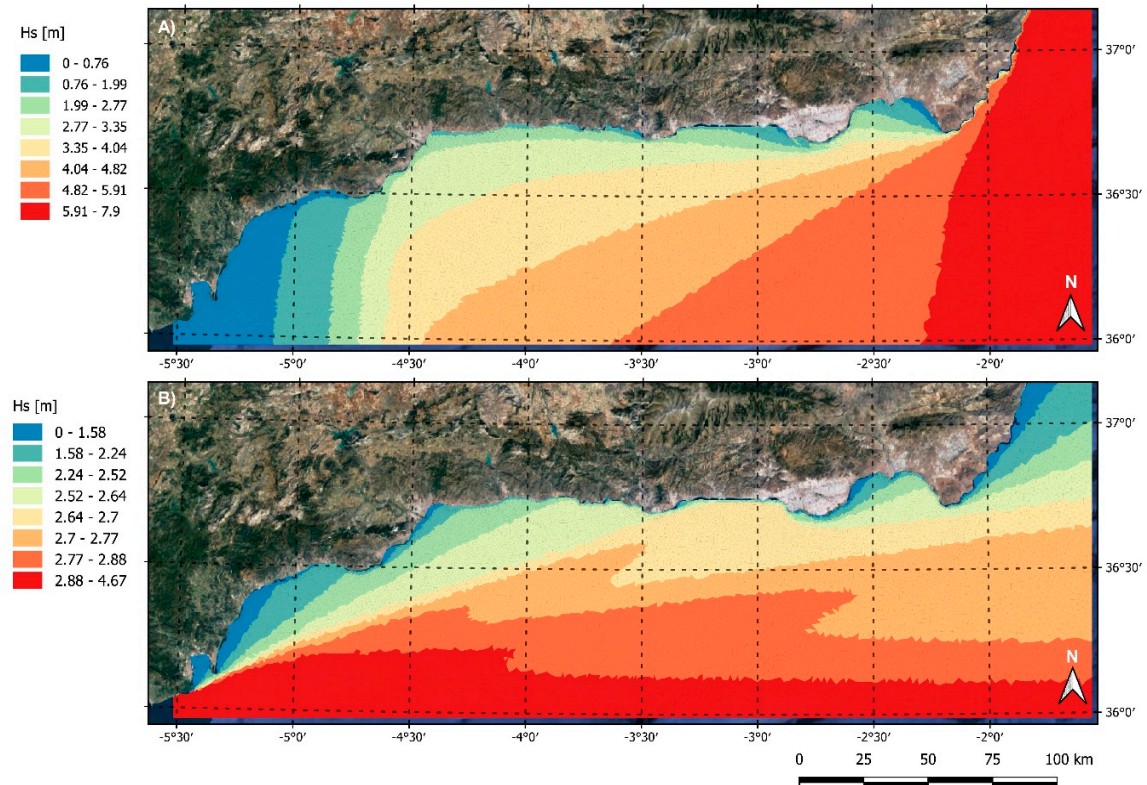

**Figure 4.** Significant wave height obtained by the numerical model run. (**A**) Offshore boundary conditions: mean wave direction = 67.5° ENE and wave height return period = 50 years. (**B**) Mean wave direction = 247.5° ENE and wave height return period = 50 years. Wave height values were divided into 8 intensity classes using the quantile method.

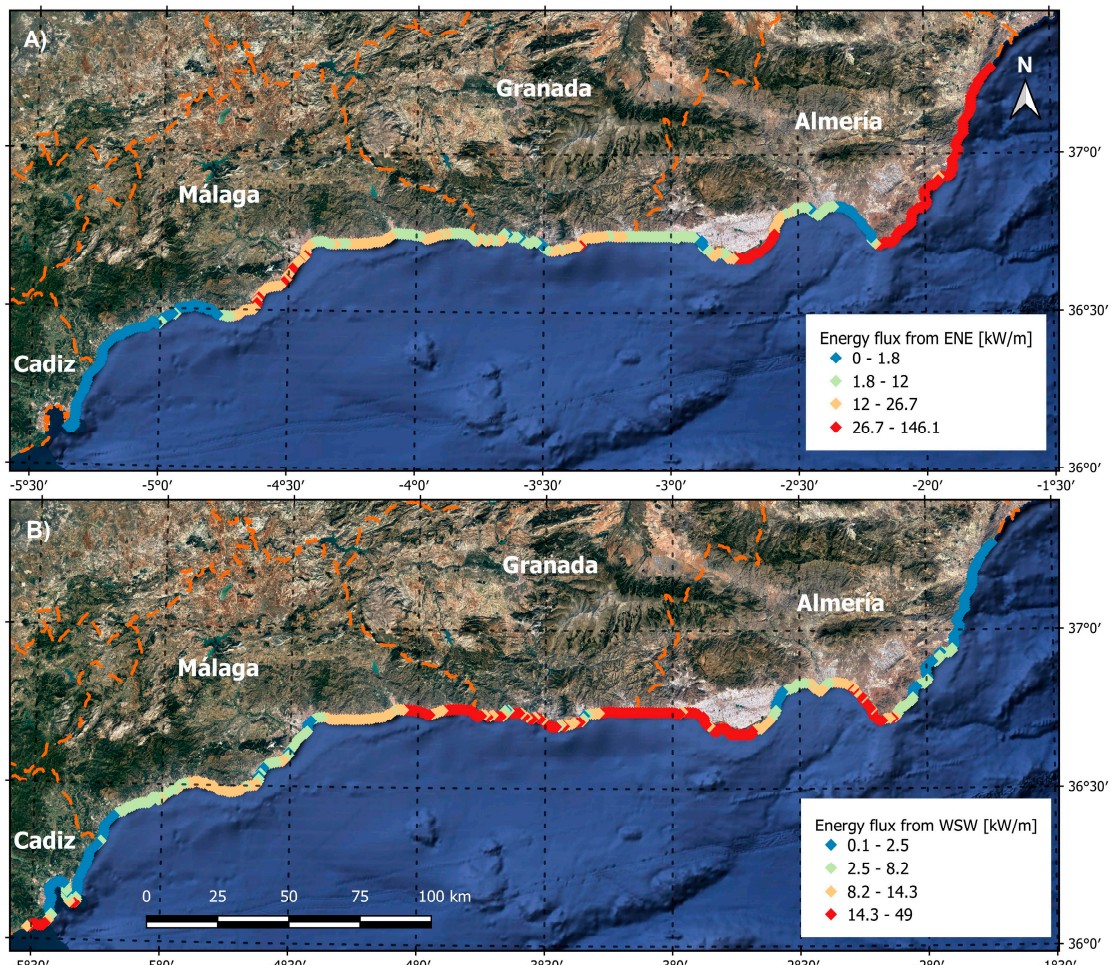

**Figure 5.** Energy flux distribution along the Andalusia coastline. Offshore boundary conditions: (**A**) mean wave direction = 67.5° ENE and wave height return period = 50 years (Figure 4A); (**B**) mean wave direction = 247.5° WSW and wave height return period = 50 years (Figure 4B).

The division of the energy flux values into classes was carried out by means of the quantile method that divided classes so that the total number of data in each class was the same.

### 4.2. Buffer Zone

In this study, dry beach widths ranged from 101.46 m at El Zabal (Cadiz Province) to 3.91 m at Los Genoveses beach (Almeria Province). Mean values of dry beach width expressed as a multiple of the 20-year predicted shoreline position showed the prevalence of class 1, that is, of a dry beach width ≥5 times the 20-year predicted shoreline position, along 52.8% of the coast.

### 4.3. Coastal Sensivity

Considering the total amount of transects studied, the most frequent class was 2 (29.63% of the coastal length, or 2985 transects), followed by classes 3 (23.19%, 2336 transects), 1 (22.60%, 2277 transects), and 4 (18.64%, 1878 transects). Class 5 represented only 5.93% of the area studied or 597 transects (Figure 6).

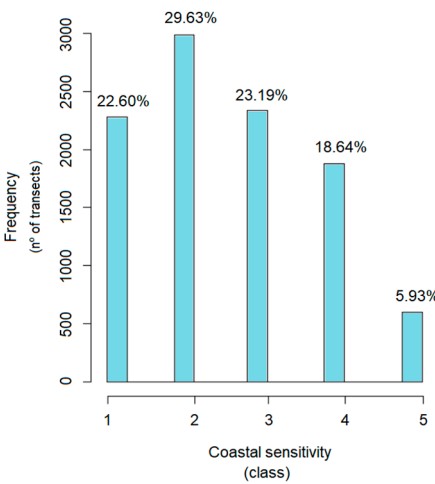

**Figure 6.** Frequency of coastal sensitivity classes.

The highest values of sensitivity were recorded at the easternmost coast of Almería Province (Los Genoveses beach, and Carboneras, La Parata, and Garrucha areas) likely due do the great exposure to wave energy (Figure 4) and the intermediate/narrow values of dry beach width. The areas with lower sensitivity values were located in the westernmost area of Andalusia, namely at Cádiz Province, from La Línea de la Concepción to Torreguadiaro, essentially because of the very low energetic conditions of this coastal sector (Figure 5) and the presence of relatively large buffer zones. The Costa del Sol area (Málaga Province), from Fuengirola to Málaga, showed low sensitivity related to low and very low energy levels (Figure 5) and the presence of a relatively wide buffer zone observed in correspondence with coastal protection structures that favored, together with the accomplishment of nourishment works [10], the formation of a stable, wide beach.

*4.4. Mitigation or Response Strategies*

The three main recommended response strategies to counteract coastal erosion processes were obtained by combining coastal sensitivity classes and land uses (Table 5) and are presented in Figure 7. In the case in which a coastal area with a low economic value (e.g., agricultural areas, etc., in Table 3, corresponding to land use classes "A" and "B" in Table 5) presented low sensitivity (i.e., classes 1 to 3 in Table 5), the area was not considered at risk and hence no action was required. In the case in which a coastal area with a very relevant economic value (i.e., extended urban and industrial areas, etc., in Table 3, corresponding to land use classes "D" and "E") presented high sensitivity (i.e., classes 4 and 5), the area was considered at the highest risk and hence protection measures were mandatory. All situations in between the two mentioned cases were considered at a medium level of risk and hence likely needed some kind of action; further studies are required in order to decide the necessary sound actions depending on the specific typology of the menaced infrastructure and level of sensitivity.

**Table 5.** Combination of coastal sensitivity and land uses to obtain the response strategies. Blue color corresponds to the "no action" option, yellow corresponds to the "adaptation" option, and red corresponds to the "protection" option.

|  |  | Land Use | | | | |
|---|---|---|---|---|---|---|
|  |  | A | B | C | D | E |
|  | 1 | 1A | 1B | 1C | 1D | 1E |
|  | 2 | 2A | 2B | 2C | 2D | 2E |
| Coastal Sensitivity | 3 | 3A | 3B | 3C | 3D | 3E |
|  | 4 | 4A | 4B | 4C | 4D | 4E |
|  | 5 | 5A | 5B | 5C | 5D | 5E |

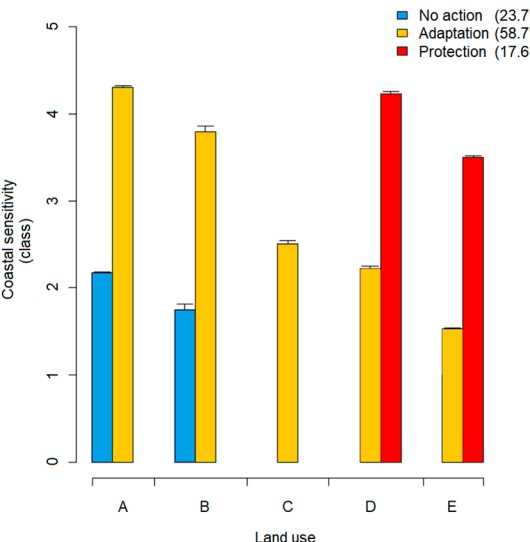

**Figure 7.** Response strategies resulting from the combination of coastal sensitivity and land use classes. Error bars are also presented. Percentages indicate the total amount of coastline classified within each response option.

The most common response strategy was "adaptation" (58.7%) followed by the "no action" (23.7%) and the "protection" (17.6%) options (Figure 8).

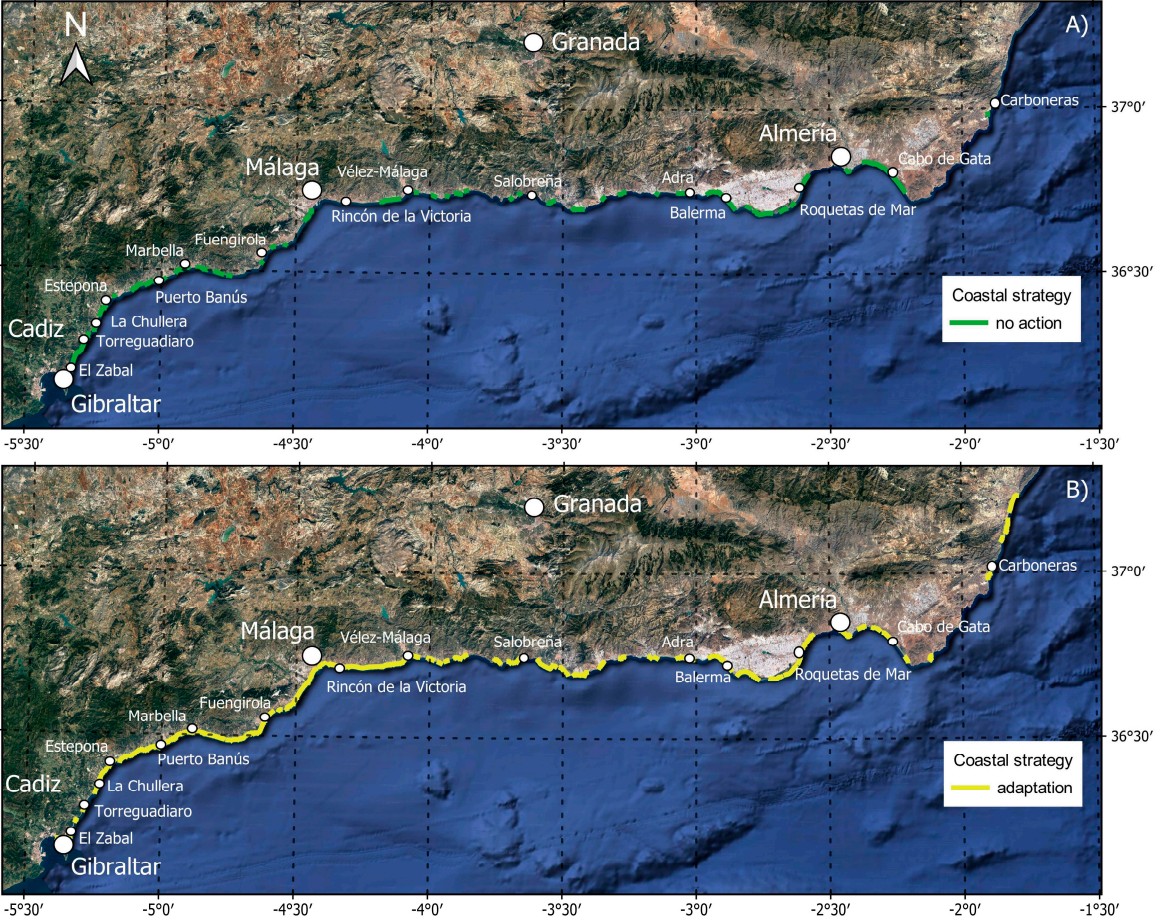

**Figure 8.** *Cont.*

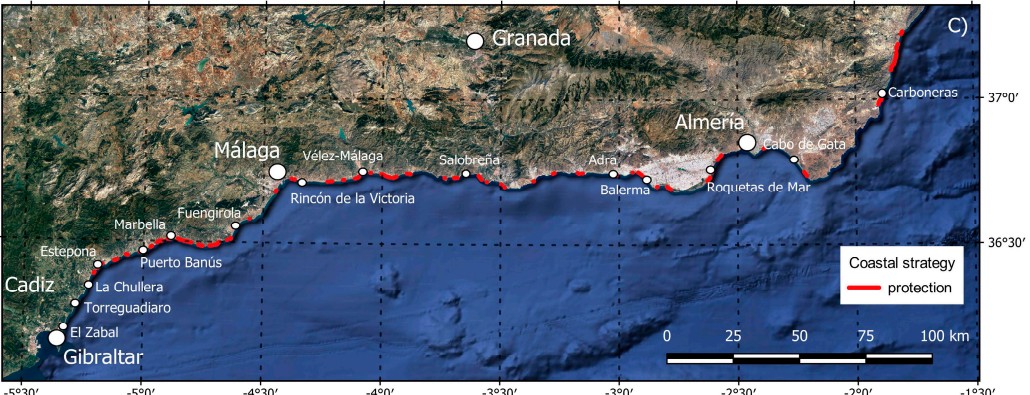

**Figure 8.** Distribution of response strategies: "do nothing" (**A**), "adaptation" (**B**), and "protection" (**C**).

## 5. Discussion

### 5.1. Wave Forcing

Rangel-Buitrago and Anfuso [9] considered wave height, storm surge, degree of littoral exposition to wave fronts [62], and tidal range [63] as forcing agents, following several investigations on storm-induced erosion [8,11,24,64–67]. In this paper, it was not possible to calculate the storm surge, and the tidal range was uniform along the area investigated; therefore, the forcing agent was represented by the wave energy flux distribution along the coast associated with very energetic conditions (i.e., a storm with a return period of 50 years). Wave energy flux distribution adequately represents forcing agents since it is the output of a wave propagation process and hence reflects several cumulative factors determining the way in which wave energy is distributed along the coastline (e.g., the degree of littoral exposure and the local bathymetric characteristics). Bathymetric conditions regulate wave shoaling and dissipation that, within coastal orientation, determine the level of exposure to the most energetic approaching wave directions (Figure 5), for example, the prevalence of longshore or shore-normal transport are very relevant in beach and dune erosion [68]. Storm waves approaching from the ENE direction predominantly affect the easternmost area of the Mediterranean coast of Andalusia from Cabo de Gata to the administrative limit of the Region, as seen in Figure 3. The rest of the coast is essentially exposed to storm waves approaching from WSW. The central area of the studied coast is characterized by the alternation of westerly and easterly waves and winds, as observed for the Costa del Sol [33,69,70], which agrees with the wave forcing shown in Figure 4 and the medium/high values of energy flux shown in Figure 5.

### 5.2. Buffer Zone Characteristics

A wide dry beach represents an efficient buffer zone to storm erosion processes, but this parameter is often considered in a subjective way. In several regional and local studies, coastal sensitivity was often expressed by means of absolute values of coastal erosion rates [68,71,72], making difficult the use and application of a unique methodology since erosion rates range considerably from place to place [9,36,66]. Hence, since the dry beach protection function is strongly related to its width and local erosion recorded rates, the dry beach width was considered in this paper as a multiple of the 20-year predicted shoreline position. This methodology is thus objective and applicable to different areas where similar data are available.

As an example of previous assumptions (i.e., the subjectivity linked to the use of beach width value as an indicator of its effectiveness as buffer zone), the results presented in this paper were compared with data recorded by Molina et al. [36]. Sometimes a wide beach records relevant accretion rates, that is, there is a direct correspondence between beach width and beach accretion trend or a narrow beach is linked to an erosional trend; this was the example of El Zabal (SW part of Cádiz

Province), which presented a wide beach up to 101.4 m and belonged to the "very wide" buffer zone class ("high accretion" class in [36]), and of Los Genoveses beach (eastern part of Almería Province), which was 3.91 m in width and belonged to the "very narrow" buffer zone class ("high erosion" of the classification used in [36]). However, there are many cases in which this premise is not met, for example, at the Natural Park Punta Entinas-El Sabinar (Almería Province) where, even though the beach is very narrow, it belongs to the narrow buffer zone class in this paper and to the "high accretion" class of the classification used in [36]. The opposite is observed at Salobreña and Carboneras that presented 196.8 and 111.6 m of beach width, respectively ("very wide" buffer zone class), but because of the recorded erosion trend, they belonged to the "high erosion" class in Molina et al. [36].

### 5.3. Land Use

Land use classes broadly coincided with the ones proposed by Rangel-Buitrago and Anfuso [9], McLaughlin et al. [22], and McLaughlin and Cooper [63]. In this paper, in order to have more objective and well-established categories, available data obtained from the official web page of the Andalusia Regional Administration, based on the SIOSE Project that was a continuation of the European Project CORINE, were used. Therefore, the information was compatible with the European land cover and was based on the Hierarchical INSPIRE Land Use Classification System (HILUCS) land use classification from the European Union's INSPIRE Directive. Such data were easily accessible at the European and Global scales as they have been integrated into the Land Monitoring Service of the Copernicus Programme, the European Union's Earth Observation Programme.

Further investigations could be focused on obtaining the percentage of urbanized area that has been expressed according to the density of human infrastructures and which broadly corresponds to the "engineered frontage" [73,74], the "coastal construction index" [75], and several sub-indexes (e.g., "settlements", "roads", and "railway") characterized by McLaughlin and Cooper [63].

### 5.4. Mitigation or Response Strategies

The "no action" option (Figure 8) appeared where both coastal sensitivity and land use classes were low, for example, at natural areas with a wide buffer zone and low/medium coastal forcing values. This was observed at the westernmost area of Cádiz Province (El Zabal Figure 9A, Torreguadiaro, and La Chullera), and at some areas from the west coast of Almería Province (Balerma, San Agustín, Costacabana, and Cabo de Gata). In such areas, no interventions are required on the decadal time scale but caution has to be focused on possible changes of the coastal trend on the large temporal scale due to climatic change-related processes (e.g., changes in storm intensity and sea level rise [26,27,29]), even though this latter process (i.e., sea level rise) seems to be not a problem for these areas [43–45].

The "adaptation" option (Figure 8) was partially linked to the great level of urbanization and the general erosive behavior of the Andalusia Mediterranean coast, a trend well documented by several authors [10,36,76–80]. Specifically, the adaptation was found to be ideal along more than one half of the studied coast, essentially in two main cases: (1) at natural areas with high sensitivity and (2) at urbanized areas with low sensitivity values. Examples of the former case are the natural area south of Carboneras, some natural areas in La Parata, and the natural area north of Garrucha (East of Almería Province). Examples of the latter case are La Línea de la Concepción (SW part of Cádiz Province), Puerto Banús (Marbella Bay, Málaga Province Figure 9B), and in general the Costa del Sol central and eastern areas (Málaga Province). Furthermore, such areas are recording sea level rise values of a few millimeters per year [43–45], so erosion problems are expected to increase in the next decades. Since this methodology was applied at a large, regional scale, within this research it was not possible to distinguish or propose detailed adaptation strategies according to coastal sensitivity and specific land use distribution/typologies. Hence, further investigations must be carried out at a smaller spatial scale in order to recommend the best adaptation strategies ranging from the "land use change" strategy (i.e., an agricultural area can be transformed into a grazing area [32,81,82]) to the "relocation" strategy (i.e., the landward movement of a coastal road [31], as could be the case of the

main Andalusia coastal highway at Mijas and Fuengirola (Málaga Province) and the National coastal road at some sectors of Vélez-Málaga and Adra (Málaga Province)). For other locations, adaptation must be devoted to the modification of existing protection structures, as proposed by Costa and Coello [83], or their abandonment according to the "do nothing" strategy [31]. This could be the case of the Puerto Banús area and many urban beaches at Málaga protected by coastal structures that have been modified several times over the last decades [10]. Such structures need periodic maintenance and/or may require modification (e.g., lowering a breakwater or a groin), thereby reducing their impact on the landscape [38], negative environmental effects [84,85], and dangerous related currents during storms [86]. Hard protection structures can be partially removed [10] and nourishment projects implemented to enlarge or maintain the dry beach width, thus increasing the beach carrying capacity [87].

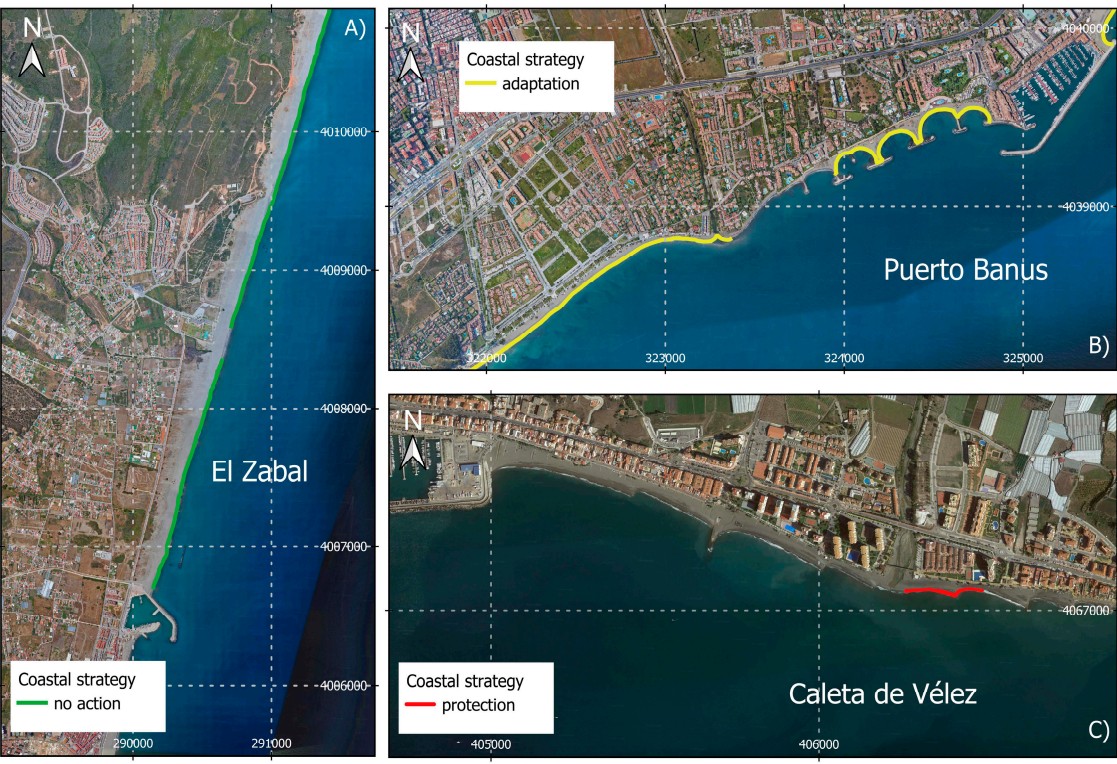

**Figure 9.** Maps of three investigated areas and associated response strategies. (**A**) El Zabal, close to La Línea de la Concepción (Cádiz Province); (**B**) Puerto Banús (Málaga Province); and (**C**) Caleta de Veléz, (Málaga Province). The projection used is the UTM30N, the referencing system is the WGS85, and the coordinates are in meters.

The "protection" option (Figure 8) characterized coastal sectors where both coastal sensitivity and land use classes presented high values, that is, at urbanized areas with a narrow buffer zone and medium/high coastal forcing values. This was observed at La Gaspara (SW part of Cádiz Province), Marbella and Mijas (center of Málaga Province), Caleta de Vélez (eastern part of Málaga Province, Figure 9C), and Balerma (east of Almería Gulf) areas that have narrow buffer zones and medium coastal forcing values, as well as at the Carboneras, La Parata, and Garrucha areas (eastern part of Almería Province) that have from medium to narrow buffer zones and high coastal forcing values. In such areas, the implementation of coastal mitigation strategies will be required in future years, and the determination of sound defense modalities is linked to coastal trend, characteristics, and human land uses.

**Author Contributions:** Data curation, R.M., G.M. and C.L.R.; Formal analysis, R.M., G.M., C.L.R. and G.A.; Methodology, R.M., G.M., C.L.R., G.A.; Software, C.L.R. and G.M.; Supervision, G.A., G.C.; Writing—original

draft, R.M., G.M., C.L.R., G.A.; Writing—review & editing, C.L.R., G.M., G.A. All authors have read and agreed to the published version of the manuscript.

**Funding:** This research received no external funding.

**Acknowledgments:** This work is a contribution to the Andalusia Research Group PAI RNM-328 and to the PROPLAYAS network. Special thanks go to Reviewer 1 whose comments greatly improved the quality of the manuscript.

**Conflicts of Interest:** The authors declare no conflict of interest.

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
