# Peer review of "A Methodological Approach to Determine Sound Response Modalities to Coastal Erosion Processes in Mediterranean Andalusia (Spain)"

_jmse, doi:10.3390/jmse8030154_

Round 1
Reviewer 1 Report
Please see the document comments.
Overall, I found this paper quite interesting. I would like to see a bit of space dedicated to sea level rise and how that will fit into the management strategies. The maps need to be a bit more consistent and usable. While I understand the regional scale in nature, the way the maps are presented makes them somewhat unusable. Perhaps a 4-panel callout with the areas of interest would help drive home the point the authors are intending to make. I would also like to see some discussion of the slope/elevation of the sandy beach. While a wide beach does provide a buffer, a narrow steep beach may also be fairly stable depending on the elevation of the back beach. In the maps, I would like to see more attention paid to the category breaks and consistent symbology so maps can be compared and assessed objectively. Figure 7 needs a bit more discussion on how the methods were used to draw these conclusions. I would like to see more discussion of the protection options and more clarification on what is accommodation and what is protected. In the paper, it discusses beach nourishment in both sections.
Overall, I found this paper quite well researched and written. There are a few places I noted that the writing needs to be addressed, particularly with verb tense. this needs to be consistent throughout.

Author Response
RESPONSE TO REVISOR 1
Dear Editor
It is a pleasure for me to inform you that we carried out all changes proposed by the two reviewers, all new added or corrected text is in this blue colour. We specially want to thank the Reviewer 1 that deeply revised the document, corrected several grammatical errors and made very interesting suggestions and observations that allowed to greatly improve the first draft of the manuscript.
Response to Reviewer 1.
First of all I want to inform the reviewer that we decided to change the term “do nothing” for “no action” because the former can give rise to confusion since it is usually used when infrastructures are at risk but their protection is very expensive and hence nothing is done to protect them, and this was not the case of our study. The “no action” is referred to essentially natural areas or low economic land uses that are at null or very low risk.
Question:
Overall, I found this paper quite interesting. I would like to see a bit of space dedicated to sea level rise and how that will fit into the management strategies.
Answer:
We clarified that the investigation proposed in this paper deals with future changes in next decades (say 2-3 decades since we used the 20-yr predicted shoreline position to evaluate future coastal behaviour) and hence, at this time scale, we consider sea level variations of low or almost null importance. Further, we added information about sea level trend in the investigated area.
Question:
The maps need to be a bit more consistent and usable. While I understand the regional scale in nature, the way the maps are presented makes them somewhat unusable. Perhaps a 4-panel callout with the areas of interest would help drive home the point the authors are intending to make.
Answer:
We modified the figure with the response strategies enlarging it and making 3 different cases and not 2 as in the previous version. We added a new figure with three detailed maps (the “zooms” required by the reviewer) regarding each one of the proposed mitigation strategies.
Question:
I would also like to see some discussion of the slope/elevation of the sandy beach. While a wide beach does provide a buffer, a narrow steep beach may also be fairly stable depending on the elevation of the back beach.
Answer:
It is indeed a very interesting point and we add some explanation in the study area and in the methodology and we added references. Unfortunately we have no detailed data on beach slope and morphodynamic state so we preferred to use beach width. Anyway beach state conditioned coastal evolution and, in order to determine coastal susceptibility, we used the 20-yr predicted shoreline position that is based on observed coastal erosion/accretion processes, so we partially included it in an “indirect” way.
Question:
I would like to see more attention paid to the category breaks and consistent symbology so maps can be compared and assessed objectively.
Answer:
We explained the used methology, limits are not intuitively designed but calculated by means of computer’s programs – details have been added in the text.
Question:
Figure 7 needs a bit more discussion on how the methods were used to draw these conclusions. I would like to see more discussion of the protection options and more clarification on what is accommodation and what is protected. In the paper, it discusses beach nourishment in both sections.
Answer:
Yes, we added an explanation about these issues; we hope it is clearer now. Yes the “adaptation” strategy includes several modalities of response. The main idea of this paper is to give a first approximation at regional scale of coastal trend and susceptibility and consequent required response options, i.e. it is a tool for Regional Administrators to analyse wide coastal areas, than detailed studies have to be carried out at smaller spatial scale to decide sound specific response strategies according to the different parameters that have to be considered (coastal trend, characteristics of the buffer zone and typologies of infrastructures). Yes nourishment was considered within both the “adaptation” and “protection” strategies since, within the former, nourishment is considered as a modification of already existing hard protection structures, e.g. the total or partial removing of an hard structure that is “replaced” by a nourishment work, meanwhile in the latter option (“protection”), nourishment is considered as a protection strategy for an area that is not yet protected.
Question:
Overall, I found this paper quite well researched and written. There are a few places I noted that the writing needs to be addressed, particularly with verb tense. this needs to be consistent throughout.
Answer:
Thank you very much we greatly appreciated your efforts to improve the quality of the paper and we did our best to carry out all your suggestions.
Reviewer 2 Report
The authors proposed an interesting topic presenting a coastal risk assessment in Andalusia.
The work was thoroughly written, the authors carried out detailed analyses, the results are very
interesting as well as the subject of the topic under consideration.
This paper is strongly recommended for publication.
Only some little amendments are needed for a suitable paper improvement.
-Several studies are available in the Mediterranean Sea in order to assess coastal risk in microtidal environment, suggesting indicators and methodology, for example:
DI RISIO, Marcello, et al. Comparative analysis of coastal flooding vulnerability and hazard assessment at national scale. Journal of Marine Science and Engineering, 2017, 5.4: 51.
ARCHETTI, Renata, et al. Innovative strategies, monitoring and analysis of the coastal erosion risk: The STIMARE project. In: The 29th International Ocean and Polar Engineering Conference. International Society of Offshore and Polar Engineers, 2019.
PASQUALI, D., et al. A simplified hindcast method for the estimation of extreme storm surge events in semi-enclosed basins. Applied Ocean Research, 2019, 85: 45-52.
-Please add some info about the mesh (minimum and maximum size) for the wave model
-Please clarify where are the points extracted for energy flux analysis? At what depth along the coastline? it's unclear for me.
-The choice of the mitigation strategy (table 4), although correct (in my opinion), seems to be very subjective. I think it would be appropriate to explain to better explain and justify your choices.
Author Response
Response to Reviewer 2
We carried out all suggestions since we cited two articles among ones suggested by the reviewer and we clarified the part about the wave model propagation adding further information in the text.